# Maternal knowledge and practice of safe infant sleep position in South Ethiopia: Implications for preventing sleep-related infant deaths

Tamene Demissie Lachore[1], Tamirat Toma Bekele[2], Alemayehu Beharu Tekle[2]*,
Nega Degefa[3], Genet Assefa[3]

**1** School of Nursing, Wolaita Sodo University, Wolaita Sodo, Ethiopia, **2** School of Medicine, Wolaita Sodo University, Wolaita Sodo, Ethiopia, **3** School of Nursing Arbaminch University, Arbaminch, Ethiopia

☯ These authors contributed equally to this work.
* alemayehub.tekle@wsu.edu.et

## Abstract

### Background

Safe infant sleep position practice is crucial for infants from birth to 12 months of age. One of the most important interventions in infancy is choosing the right sleeping position for the baby. The unsafe position of the child while sleeping is associated with sudden infant death Syndrome. However, little is known about the safe infant sleep position practice in Ethiopia. Therefore, this study aimed to assess safe infant sleep position practices among mothers attending child health care services in the Wolaita Zone public hospitals, South Ethiopia, 2024.

### Methods

The study included 569 mothers with infants attending child health care services in public hospitals in the Wolaita Zone. Participants were selected using a systematic sampling method, and data was collected through structured questionnaires administered via interviews. The collected data was checked for completeness, coded, and entered into EpiData version 4.6. Analysis was conducted using the social science statistical package. Descriptive statistics were used to summarize the data, and results were presented through narratives, tables, and figures. Binary logistic regression analysis was employed to identify explanatory variables associated with safe infant sleep position practices. A p-value of ≤0.05 was considered statistically significant.

### Results

A total of 569 study participants were included in this study with a response rate of 99.1%. The safe infant sleeping position practice (back to sleep) in this study was

**Data availability statement:** All relevant data are within the paper and its Supporting Information files.

**Funding:** The author(s) received no specific funding for this work.

**Competing interests:** The authors have declared that no competing interests exist.

**Abbreviations:** AAP, American Academy of Pediatrics; AMU, Arba Minch University; ANC, Antenatal Care; AOD, Adjusted Odds Ratio; BSC, Bachelor of Science; CDC, The Centers for Disease Control and Prevention; CI, Confidence Interval; CSA, Central Statistical Agency of Ethiopia; ETB, Ethiopian Birr; FGD, Focused Group Discussion; GERD, Gastro esophageal Reflux; MCH, Maternal and Child Health; SIDS, Sudden Infant Death Syndrome; SUDI, Sudden Unexplained Death of Infants; UK, United Kingdom; US, United States; WHO, World Health Organization; WSUCSH, Wolaita Sodo University Comprehensive Specialized Hospital.

42.7% [95% CI (38.6%, 46.8)]. Maternal occupation, parity, source information about sleep position, and good knowledge of safe sleep position practice were significantly associated by (AOR = 3.49; 95% CI (1.81, 6.76)), (AOR = 0.39; 95% CI (0.21, 0.74)), (AOR = 4.64; 95% CI (1.77, 12.18)), and (AOR = 2.04; 95% CI (1.22, 3.40)) respectively.

## Conclusion

The study found that over half of participants did not practice safe infant sleeping positions. Factors such as maternal occupation, parity, information sources, and knowledge significantly influenced safe sleep practices. Targeted health education—particularly through antenatal and postnatal care services, as well as community-based awareness campaigns—is recommended to improve maternal knowledge and reduce sleep-related infant deaths.

## Introduction

The practice of safe sleep position for infants is a critical aspect of infant care. Safe sleep position refers to placing infants in a specific position during sleep to reduce the risk of sudden infant death syndrome (SIDS) and other sleep-related incidents. It is recommended that all caregivers allow infants up to 1 year of age to sleep on their backs whenever they sleep. Side sleeping is unsafe and not recommended. Lying supine on a flat, non-inclined surface does not increase the risk of choking and aspiration in infants and is recommended for all sleep, including for infants with gastro-esophageal reflux disease (GERD) [1,2].

Historically, there have been significant shifts in recommendations regarding infant sleep positions. In the past, placing infants on their stomachs was a common practice. However, research and data analysis conducted over years have provided valuable insights into the relationship between sleep position and infant safety [3].

In the early 1990s, a groundbreaking study known as the "Back to Sleep" campaign (now called "Safe to Sleep") was conducted by the American Academy of Pediatrics (AAP) and other organizations. This study demonstrated a strong association between placing infants on their backs to sleep and a reduced risk of SIDS. The campaign aimed to raise awareness among parents and caregivers about the importance of placing infants on their backs for sleep [3–6].

Since the launch of the campaign, the incidence of SIDS has significantly declined worldwide. Numerous studies have confirmed the effectiveness of back sleeping in reducing the risk of SIDS. Placing infants on their backs to sleep has become the recommended sleep position by major health organizations, including the AAP, the Centers for Disease Control and Prevention (CDC), and the World Health Organization (WHO) [7,8].

The rationale behind the recommendation for back sleeping is that it helps maintain an open airway for infants, reducing the risk of suffocation or overheating. It is

important to note that side sleeping is not considered as safe as back sleeping, as infants can inadvertently roll onto their stomachs while in this position [1,4].

While back sleeping is the preferred position, other safe sleep practices should also be followed. These include using a firm sleep surface, such as a crib or bassinet with a fitted sheet, ensuring that there are no loose or soft bedding items present (e.g., pillows, blankets, bumper pads), and keeping the sleeping environment at a comfortable temperature [1].

Safe sleep position practice was identified as a critical intervention to prevent SIDS. Infant death rates from SIDS decreased in several countries following the success of international public health initiatives advising on safe baby sleep position practices [9,10]. As a result of a back- to- sleep campaign, the rate of SIDS in the United States decreased by 53% between 1992 and 2001 [3].

Each year infants die of unsafe sleep practices. Placing infants in the supine position for sleep reduces the risk of dying, yet infants continue to be placed in other than the supine position for sleep [11]. An unsafe sleeping position practice in infants is associated with SIDS and other sleep-related deaths [4,12,13]. SIDS is nearly 2,500 death per year in the United States; in the UK, the incidence of SIDS was ± 0.27/1,000 live births, while in Italy it was around 1/1,000 live births [14,15]. Likewise, the prevalence of SIDS points to high rates in Africa. A systematic review found the range to be 3.7/1,000 live births in South Africa, 2.5/1,000 live births in Niger, and 0.2/1,000 live births in Zimbabwe [16].

Infants placed in a safe sleep position were less likely to die due to SIDS cases compared to unsafe sleep position practices. Among, majority of infants were reported to have been placed in unsafe positions prior to death. The SIDS case rate in the lateral position was 53% to 64%, whereas 23.6% to 37% of SIDS cases were placed in the prone position compared to 10% to 12% in the supine or back position [16–18].

An unsafe sleeping position practice in infants is associated with SIDS and other sleep-related deaths [4,12,13]. Infant deaths related to unsafe sleep position were decreased in many countries following the safe-to-sleep position recommendation [9,10]. Albeit there were many studies conducted in Ethiopia regarding infant mortality, no findings show infant sleep-related deaths [19]. Additionally, only one study identified that safe infant sleep position practice is very low in Ethiopia [20]. In Ethiopia, understanding the practice of safe infant sleep position is crucial due to the high prevalence of infant mortality. Identifying the factors associated with safe sleep practices can help healthcare providers and policymakers develop targeted interventions to improve maternal knowledge and promote safe infant sleep practices [19].

Safe infant sleep position practice is very low in many countries. A descriptive study conducted in Turkey found that 47.5% of parents reported placing their child to sleep in the supine position, 43.6% in the lateral position, 2.5% in the recumbent position, and 6.4% in two or three positions. Also, 9.8% of participants share the infant's bed, and 92.9% of them share a room [12]. Likewise, a cross-sectional study conducted in Riyadh, Saudi Arabia, found that 12.4% babies slept on their stomachs, 58.6% on their backs, and 28.9% on their sides. While 10.8% of mothers who had infants who slept in a separate room and 42% indicated that the child had shared a bed with a parent [21]. Additionally, another cross-sectional study was conducted in Family Health Centers (FHC) in Turkey and found that 36.9% preferred the supine position for their child. In addition, 5.4% of parents and babies shared a room. 6.8% of parents and babies shared a bed [22]. On the other hand, in a study conducted in Brazil, the average prevalence of children who regularly slept on their backs was 55.4%, And 53.2% did not report bed sharing [10].

Findings from Africa also revealed very low application of safe infant sleep position practices. A study conducted in Nigeria found that infant sleep position practice on the back (supine) was 18.1%, and 71.7% of infants share a bed with parents or siblings [23]. In a cross-sectional survey of mothers in Lusaka, Zambia, in 2022, only 6.7% of parents reported placing their babies to sleep on their backs [24]. And the recommended practice of sleeping position (supine) in Ethiopia was 33.5% [20].

In general, systematic reviews in Africa found that a minority (2.7% to 21.5%) of mothers placed the infants in the supine position during sleep, and bed sharing was reported at a rate of 60% to 91.8% among mothers of infants at well-baby clinics [16].

To reverse infant mortality, Ethiopia has made significant investments in basic healthcare over the previous decade, focusing mostly on the poor rural population. Attendance at antenatal care (ANC), access to primary health care, coverage of fully vaccinated children, and competent deliveries have all improved dramatically, with maternal health services now provided free of charge at all health institutions. As a result, a considerable decrease in newborn and child mortality has been seen [25]. Beyond these, there was a missed opportunity to address infant sleep safety.

One of the most significant interventions in early life is selecting the appropriate sleep position. A well-known health promotion method that improves adherence to safe practices and reduces mortality is identifying newborn sleeping position habits and offering information to educate the community [22,26]. However, there is a single study in Ethiopia and no study in my study area on the degree of adherence to safe sleep position practice and its contributing factors. Therefore, it is important to shed light on the safe infant sleep practices of mothers about the sleeping position of the infant.

The primary beneficiaries are the infants themselves. Safe sleep practices can significantly reduce the risk of SIDS and other sleep-related incidents. By identifying factors associated with unsafe sleep practices, interventions and educational programs can be developed to promote safer sleep environments for infants, ultimately improving their health and well-being.

Mothers play a crucial role in making decisions about their infants' sleep environment. By assessing their knowledge and practices related to safe sleep positions, interventions can be tailored to address any gaps or misconceptions. This can empower mothers with accurate information and guidelines, enabling them to make informed decisions that promote the safety of their infants during sleep.

Healthcare professionals, such as pediatricians, nurses, and midwives, can directly benefit from the findings of this assessment. Understanding the factors associated with safe sleep practices among mothers can inform healthcare providers' guidance and recommendations. It can help them identify areas where additional support or education is needed and allow them to provide targeted advice to mothers regarding safe sleep positions for their infants.

The findings are important for researchers and policymakers through the contribution of the existing body of knowledge on safe sleep practices and associated factors. They can inform the development of evidence-based guidelines and policies aimed at promoting safe sleep practices among mothers and reducing the incidence of sleep-related incidents in infants.

So this study aims to assess safe infant sleep position practices among mothers and their associated factors in Wolaita Zone public hospitals, South Ethiopia, 2023.

This proposed study answered these core research questions:

• What percentage of mothers practice safe sleep positions for their infants?

• How do socio-demographic factors, service – and professional- related factors, and prior knowledge affects the safe infant sleep position practices?

## Method and materials

### Study area

The study was conducted in public hospitals in the Wolaita zone. The Wolaita area is 327 km south of Addis Ababa. Data on the demographics, education, health, and infrastructure of the Wolaita Zone in 2018: the total population of the territory is 6,142,063 people. Of the total population of the region, 3,115,050 are women and 3,027,013 are men. There is one fully specialized hospital and eight primary public hospitals in the zone [27].

### Study design and period

An institutional-based cross-sectional study design was employed in selected public hospitals of the Wolaita Zone from January 05 to February 4, 2024.

## Population

**Source population.** All women who have infants less than 1 year old and were attended by child health care services in Wolaita Zone hospitals are the source population.

**Study population.** Mothers who have infants less than 1 year old and attended child health care services in Wolaita Zone selected hospitals during the data collection period were the study population.

**Study unit.** Individual woman-infant pair who attended Wolaita Zone hospitals during data collection.

## Inclusion and exclusion criteria

### Inclusion criteria.

• Mothers with infants less than 12 months of age presenting to the public hospitals of the Wolaita Zone during the study period.

### Exclusion criteria.

• Infants presented to the clinic by non-caregivers/not bringing up the infant.

• Mothers presented with critically ill infants.

• Children with congenital conditions like micrognathia, specific to certain positions(This exclusion was made to reduce selection bias, as the preferred position of an infant with micrognathia is prone or side sleeping to reduce the risk of upper airway obstruction).

• Infants whose mothers are health care workers. (*This exclusion was made to reduce knowledge-related bias, as health-care professionals are more likely to have formal training or exposure to guidelines regarding safe sleep practices. Their inclusion could overestimate the general population's knowledge and behavior.*)

## Sample size determination and sampling procedure

**Sample size determination.** The required sample size for the first specific objective is calculated by using the single population proportion formula with the assumption of a 95% confidence level, a 5% margin of error, and a proportion of mothers' practice on infant sleep position of 33.5% from a study done in Jimma town public health institutions, Ethiopia, in 2022 [20].

$$n = \frac{(Z_\alpha/2)^2 \, p(1-p)}{d^2}$$

Where n = sample size/the desired sample size)
   Zα/2 = Standard (1.96)
   P = proportion of infant sleeping position practice = 33.5% = 0.335
   d = margin of error = 5% = 0.05

$$\text{Therefore, } n = \frac{(1.96)^2 \, 0.335(1-0.335)}{(0.05)^2} = 342.3 = 343$$

Considering a non- response rate of 10%, the sample size became 377.

Table 1 below shows the calculated sample size for associated factors to infant sleep position practice.

Hence the sample size of sleep position knowledge (574) was taken because it is the largest of all sample sizes.

**Table 1. Calculated sample size for associated factors to infant sleep position practice among mothers in Wolaita Zone public hospitals in 2023.**

| Factors(variables) | CI | Power | % of outcome among exposed | % of outcome among unexposed | N | Adding NR rate of 10% | Reference |
|---|---|---|---|---|---|---|---|
| Marital status | 95% | 80 | 84.6% | 64.59% | 166 | 183 | [20] |
| Residence | 95% | 80 | 76.9% | 63.5% | 394 | 434 | [20] |
| Sleep position knowledge | 95% | 80 | 75 | 63.3 | 522 | 574 | [20] |
| Gravidity | 95% | 80 | 72.6 | 59.67 | 450 | 495 | [20] |

## Sampling procedure

There are 9 public hospitals (1 tertiary hospital and 8 primary hospitals) in the Wolaita Zone. Three hospitals were selected by random sampling method, and then proportional allocation was done for each hospital in terms of the previous year's the same monthly report number of child care services to the data collection period. A systematic sampling technique was used to select study participants who were attending child care services during the data collection period. The total number of the previous monthly report of child care services in selected public health care institutions was taken as a total population. Then, to obtain the sampling interval (k-value), the formula k = N/n was used. Where k is the constant value, N is the previous year's monthly report number of child care services, and n is the sample size. The first study subject was selected by the lottery method. Data were collected for every 6th woman who attends child care services in the selected hospitals (Fig 1).

## Data collection methods

**Data collection instrument and data collectors.** Data were collected through face-to-face interviews using a structured and pre-tested questionnaire. The questionnaire was initially prepared in the English language and then translated into the local language (Wolatic). The tool covers sociodemographic characteristics, obstetric characteristics, women's knowledge of sleeping positions, sources of information, and sleeping position practice (annexed separately). Three nurses with bachelor's degrees collected the data, and three nurses with masters of nursing supervised the data collection process. The principal investigator has supervised and monitored the overall data collection process.

**Data collection procedure.** Prior to the actual data collection, the purpose of the study was clearly explained to the participants, and their consent was obtained. The mothers were interviewed after they received the service in a relatively quiet place separately.

**Data quality control.** The qualities of data were assured before data collection, during data collection, and after data collection. Prior to the data collection, the questionnaire was pre-tested on 5% (24) of the sample size. The reliability of data were checked by Cronbach's alpha, which was 0.741. And one day training was given for data collectors and supervisors. During data collection, close follow-up was done by supervisors and the principal investigator. After data collection, data were checked for completeness and correctness on the spot.

## Study variables

**Dependent variables.** The safe infant sleeping position practice

**Independent variables. Socio-demographic-related characteristics**: age, ethnicity, educational status of women, educational status of husband, marital status, residency, occupation, infant's age, and income.

**Obstetric-related characteristics**: parity, gravidity, ANC visit, and place of delivery.

**Sources of information**: grandmothers, health professionals, books, social media, neighbors, and friends.

**Operational definition and measurement. Maternal Infant Sleep Position Knowledge:** Maternal knowledge of infant sleep practice will be assessed through 10 knowledge assessment questions, which are adopted in English from

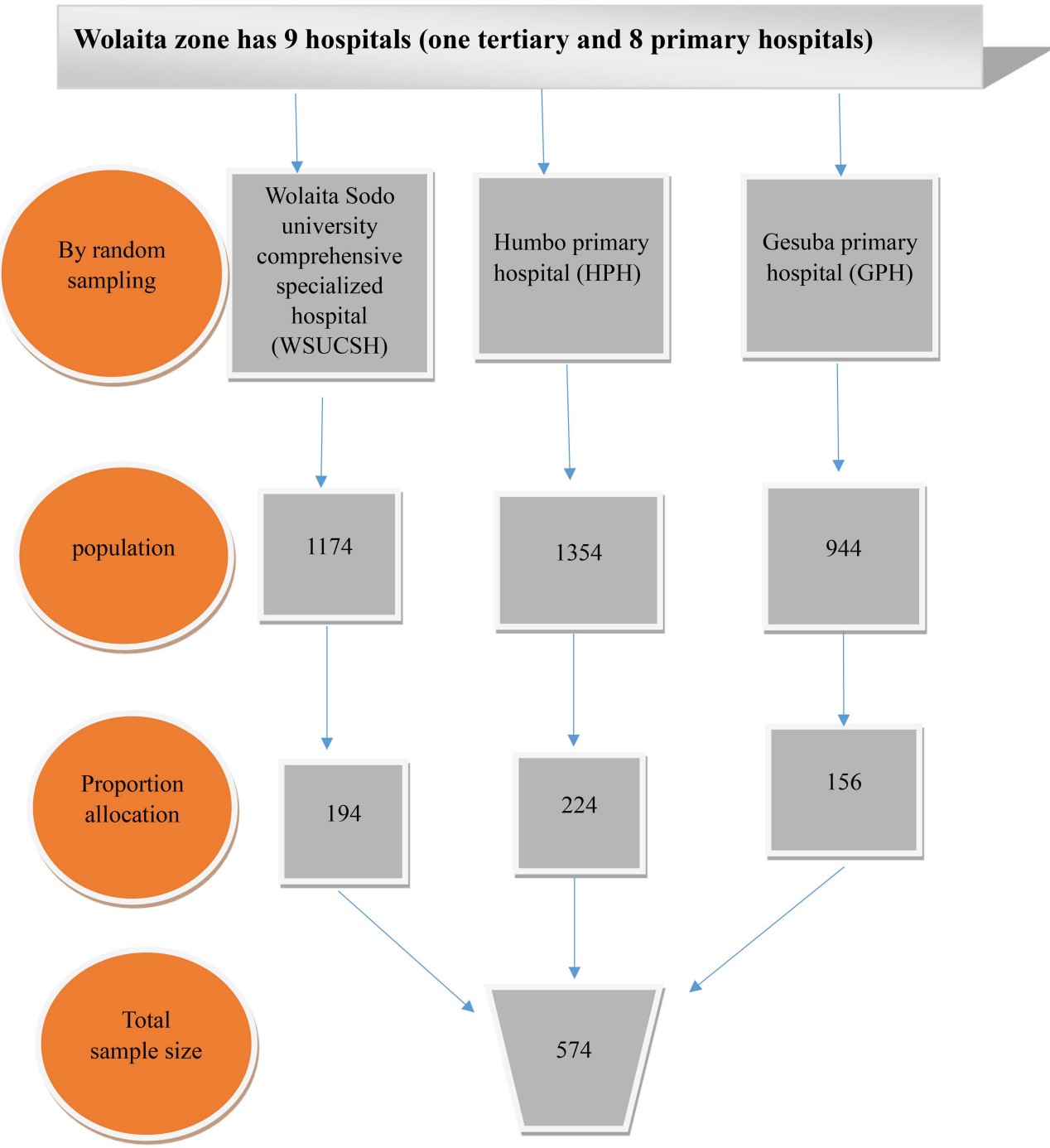

**Fig 1. Sampling procedure.**

different published literature [13,21,28]. Each question responded to with the right answer will allot a score of one, and every wrong answer will be given a score of zero. The total attainable score in the knowledge questionnaire is 10.
**Good Knowledge:** Respondents who scored **5**0% or above (≥5 out of 10 questions) were categorized as having *good knowledge*. This pre-set cut-off has been used in similar studies and reflects a meaningful level of understanding. [20].

**Poor knowledge:** Respondents who scored below 50% (<5 out of 10 questions) were categorized as having poor knowledge [20].

**Safe Infant Sleep Position Practice:** placing infants on their backs to sleep for every sleep time, including naps and nighttime sleep [1,4,29].

**Knowledge about SIDS:** Mothers who had heard about SIDS, knowledge of infant death occurring before 1 year caused by exogenic triggers/stressors (unsafe sleep position), critical period of development/immature cardiorespiratory that leads to failure of protective response [1,3,4].

### Data analysis

The collected data were checked for completeness, coded, and entered into EpiData version 4.6, and then exported to SPSS version 25 for statistical analysis. Descriptive statistics (frequency, percentage, mean, and standard deviation) were used to summarize sociodemographic, obstetric, knowledge, and practice-related variables. Results were presented in tables, narratives, and figures. To identify factors associated with safe infant sleep position practices, binary logistic regression was first used for bivariate analysis. Variables with a p-value ≤ 0.25 were selected as candidate variables for multivariable logistic regression analysis. Before running the multivariable model, we checked for multicollinearity among independent variables using Variance Inflation Factor (VIF) and tolerance tests. No significant multicollinearity was detected, as all VIF values were below the accepted cutoff of 10, with the maximum VIF observed being 2.9. The strength of association between independent variables and safe sleep practice was determined using adjusted odds ratios (AORs) with 95% confidence intervals (CIs). A p-value ≤ 0.05 was considered statistically significant in the final model.

### Ethical consideration

Ethical clearance was obtained from the Research and Ethical Review Board of Arba Minch University, College of Medicine and Health Sciences, under protocol number **TD23120**. A formal letter of cooperation was submitted to the administrative bodies of the selected hospitals to facilitate the data collection process.

Before participation, all eligible mothers were informed about the purpose, procedures, potential risks, and benefits of the study. They were assured of confidentiality, anonymity, and their right to decline or withdraw from the study at any time without affecting their access to healthcare services.

For **literate participants**, written informed consent was obtained by signing a consent form.

For **illiterate participants**, the following procedure was used to ensure ethical and autonomous participation:

- The informed consent form was read aloud to the participant in the local language (Wolaitigna).

- All questions were answered clearly, and understanding was confirmed through verbal feedback.

- After confirming verbal agreement to participate, the participant was asked to provide **a thumbprint** on the consent form as an alternative to a signature.

- A literate witness, independent of the research team and selected by the participant (e.g., accompanying relative or community member), signed the form to verify the process.

All personal identifiers were removed from the data collection tools to maintain participant confidentiality. Data were stored securely and accessed only by the principal investigator and authorized supervisors.

### Results

### Socio-demographic characteristics of the respondents

A total of 569 study participants were included in this study, with a response rate of 99.1%. Of the total participants, 155 (27.2%) were in the age group between 25 and 30 years. The mean age with SD of the mothers was 28 ± 6.9. The age

range of respondents was 23 years. Almost all, 98.9%, of participants were married. Slightly more than half (52.9%) reside in urban areas. The majority 212 (37.3%) of study participants' educational status was above secondary or higher, and the majority 245 (43.1%) were housewives. Nearly more than half of infants (52.4%) were males, and 53.8% were in the age category of 0–3 months old. The mean age of infants was 3.9 months (Table 2).

### Obstetric related characteristics of mothers

The majority of infants (352; 61.9%) in the study were not the firstborn child in the family. Three hundred fifty-nine (63.1%) of respondents were multigravida. Concerning parity Two hundred eight (36.6%) were primipara. The majority (525, or 92.3%) of respondents had ANC follow-ups during their last pregnancy, and 376 (66.1%) had more than four ANC visits. Of 569 mothers, 546 (96%) were delivered in a health institution (Table 3).

Among mothers, 67.5% heard about Sudden Infant Death Syndrome. More than half, 57.1%, didn't hear about sleep position. 136 (23.9%) of participants gained information from health professionals, and the majority (96.3%) of mothers prefer to apply advice from health professionals. The majorities (85.1%) prefer the same bed, and 90.9% use a pillow for infants (Table 4).

The majority of infants (95.8%) sleep in crib and 96.5% of infants do not sleep in separate rooms. Only 6.3% of infants are not sharing a bed with caregivers. Of 569 participants, 94.6% have no smoke exposure during pregnancy and after birth. Almost all infants (98.6%) were placed to sleep in positions other than supine. Among them, 56.1% usually slept on their side, and 1.2% in the prone position, while only 42.7% were consistently placed on their backs. (Table 5).

### Safe sleep position knowledge

Three hundred thirty-six (59.1%) participants had good safe sleep position knowledge (Fig 2).

### The safe infant sleep position practice

The safe infant sleeping position practice (back to sleep) in this study was 42.7% [95% CI (38.6% − 46.8%)] (Fig 3).

### Factors associated with safe infant sleep position practice

In bivariate logistic analysis, some potential candidate variables were selected for the multivariable logistic regression model, such as maternal age, residence, occupation, family size, family income, parity, ANC follow-ups, number of ANC visits, and source of sleep position information, SIDS information, and proper sleep position knowledge. Those variables with a p-value ≤ 0.25 were included in the variables for multivariate logistic regression analysis.

Among ten variables with p-value ≤ 0.25, four variables—maternal occupation, parity, source of sleep position information, and sleep position knowledge—were significantly associated with safe sleep position practice.

Being employed and self-employed will increase the odds of practicing a safe sleep position by 3.49 (AOR = 3.49; 95% CI (1.81, 6.76)) and 1.75 (AOR = 1.75; 95% CI (1.05, 2.91) times as compared to housewives. Compared to those grand multiparas, primiparas and multiparas had a lower likelihood of practicing by 0.39 (AOR. = 0.39; 95% CI (0.21, 0.74)) and 0.31 (AOR = 0.31; 95% CI (0.13, 0.74) times, respectively. Respondents who had heard about safe sleep positions from health professionals were 4.461 times more likely to practice than those who had not heard (AOR = 4.64; 95% CI (1.77, 12.18)). Mothers who had a good knowledge of safe practice were 2.04 times more likely to practice than their counterparts (who had poor knowledge about safe practice) (AOR = 2.04; 95% CI (1.22, 3.40)) (Table 6).

## Discussion

This study aimed to determine the prevalence and associated factors of safe infant sleep position practice among mothers attending public hospitals in Wolaita Zone, South Ethiopia. The overall prevalence of safe infant sleep position practice

**Table 2.** . Socio-demographic Characteristics of Mothers Attending Mother and Child Health Service in Wolayta zone public hospitals South Ethiopia 2023.n= (569).

| Variables | Category | Frequency | Percent |
|---|---|---|---|
| Age of mother (in years) | ≤ 18 | 51 | 9.0 |
| | 19-24 | 140 | 24.6 |
| | 25-30 | 155 | 27.2 |
| | 31-36 | 128 | 22.5 |
| | ≥37 | 95 | 16.7 |
| Marital status | Married | 563 | 98.9 |
| | Single | 6 | 1.1 |
| Residency | Rural | 268 | 47.1 |
| | Urban | 301 | 52.9 |
| Educational status of mother | No formal education | 25 | 4.4 |
| | Primary education | 183 | 32.1 |
| | Secondary education | 149 | 26.2 |
| | Higher education | 212 | 37.3 |
| Mother occupation | Employed | 121 | 21.3 |
| | Unemployed | 9 | 1.6 |
| | Self employed | 158 | 27.8 |
| | Student | 36 | 6.3 |
| | Housewife | 245 | 43.1 |
| **Monthly family income** * | Less than 1500 ETB (27.3 USD) | 103 | 18.1 |
| | 1501 −3500 ETB (27.3 USD-63.6USD) | 152 | 26.7 |
| | >3500 ETB (63.6 USD) | 314 | 55.2 |
| Family size | 1-4 | 362 | 63.6 |
| | 5-8 | 162 | 28.5 |
| | above 9 | 45 | 7.9 |
| The number of children in the house | 1-4 | 410 | 72 |
| | 5-8 | 156 | 27.5 |
| | 9 and above | 3 | .5 |
| Sex of the infant | Male | 298 | 52.4 |
| | Female | 271 | 47.6 |
| Age of the infant | 0–3 months | 306 | 53.8 |
| | 4–6 months | 182 | 32 |
| | 7 and above | 81 | 14.2 |

*Using the 2024 exchange rate 1 USD≈55 ETB

was 42.7% [95% CI: 38.6%–46.8%]. Factors significantly associated with safe sleep practices included maternal occupation, parity, source of information, and knowledge about safe infant sleep.

The prevalence found in this study was lower than that reported in studies from Saudi Arabia (61.9%, 58.6%) [28], the United States (77.3%, 78%, 95.6%) [11,30,31], Turkey (47.5%) [12], Brazil (55.4%) [32], and Iraq (54%) [33].These differences may be due to variations in methodology, study settings, socioeconomic status, and public health infrastructure. For instance, studies from the U.S. and Brazil included participants who were exposed to nationwide "Back to Sleep" campaigns and had higher access to antenatal education and health messaging. In some cases, data were collected through self-administered electronic surveys, indicating a more literate sample, which may correlate with higher health literacy. Additionally, different study designs (e.g., cohort vs. cross-sectional), larger sample sizes, and specific inclusion

**Table 3. Obstetric History of Mothers Attending MCH Service in Wolaita Zone Public Hospitals, South Ethiopia, 2023.**

| Variables | Category | Frequency | Percent |
|---|---|---|---|
| Is this your first child | Yes | 217 | 38.1 |
| | No | 352 | 61.9 |
| Gravidity | Primigravida | 210 | 36.9 |
| | Multigravida | 359 | 63.1 |
| Parity | Primipara | 208 | 36.6 |
| | Multipara | 172 | 30.2 |
| | Grand multipara | 189 | 33.2 |
| ANC During last pregnancy | Yes | 525 | 92.3 |
| | No | 44 | 7.7 |
| Number of ANC visit | No | 44 | 7.7 |
| | 1-3 | 149 | 26.2 |
| | 4 and above | 376 | 66.1 |
| Place of delivery | Home | 23 | 4 |
| | Health institution | 546 | 96 |

ANC = Antenatal care.

criteria may have contributed to higher reported prevalence in those settings. It may partly be due to the phrasing of the question ('how does the infant usually sleep?'), which reflects both maternal placement and infant movement during sleep. As infants grow older, spontaneous rolling may reduce the likelihood of maintaining a supine position despite being initially placed on their backs.

On the other hand, the prevalence found in our study was higher than that reported in a systematic review of African countries (ranging from 2.7% to 21.5%) [16], and in individual studies conducted in Turkey (28%, 22.1%, 36.9%) [13,22,34], Nigeria (29.4%) [35], and Ethiopia (33.5%) [20]. The higher figure in our study could be due to differences in sampling methods, urban-based recruitment, or increased exposure to maternal and child health services in recent years. However, caution is needed in interpretation, as variations in study period, questionnaire structure, data collection methods, and cultural beliefs may also contribute to such discrepancies.

Regarding maternal occupation, mothers employed in government or self-employed positions were more likely to practice safe sleep. This may be related to higher education levels, better access to health information, or broader exposure to professional environments where health-promoting behaviors are encouraged.

In terms of parity, our study found that primiparous and multiparous mothers were less likely to practice safe sleep compared to grand multiparas. This finding aligns with a previous Ethiopian study where multigravida women were more likely to use recommended sleep positions [20]. However, it contrasts with findings from Brazil, where increasing parity was associated with reduced adherence to safe sleep recommendations [36]. These conflicting results suggest that cultural, educational, or experiential factors related to parity may differ across settings.

Mothers who received sleep position information from healthcare professionals were significantly more likely to practice safe sleep, which is consistent with findings from the U.S. [11]. In contrast, a Brazilian study noted that mothers who relied on informal sources, such as grandmothers, were more likely to use unsafe sleep practices [36]. These findings highlight the critical role of healthcare providers in delivering evidence-based guidance to mothers, especially during antenatal and postnatal visits.

Maternal knowledge of safe infant sleep practices was another strong predictor. Mothers with good knowledge were twice as likely to practice safe sleep positioning, which aligns with previous studies in Ethiopia [20], Georgia [37], and

**Table 4. Knowledge assessment toward safe infant sleeping position (n = 569).**

| | | | |
|---|---|---|---|
| Have you ever heard about Sudden Infantile Death Syndrome (SIDS)? | Yes | 384 | 67.5 |
| | No | 185 | 32.5 |
| Have you heard about infant sleep position | Yes | 244 | 42.9 |
| | No | 325 | 57.1 |
| where did you hear it about sleep position? | My mother | 27 | 4.7 |
| | Mothers in law | 23 | 4.0 |
| | Grand mothers | 58 | 10.2 |
| | Health professionals | 136 | 23.9 |
| | I didn't hear | 325 | 57.1 |
| Advice from whom you want to apply | My mother and mothers in law | 5 | .9 |
| | Grand mothers | 16 | 2.8 |
| | Health professionals | 548 | 96.3 |
| What is preferred sleep position for infants? | Back | 232 | 40.8 |
| | Prone | 11 | 1.9 |
| | Side | 291 | 51.1 |
| | I don't know | 35 | 6.2 |
| What is appropriate environment to sleep an infant? | Separate room from parents. | 16 | 2.8 |
| | the same room with parents | 475 | 83.5 |
| | I don't know | 77 | 13.5 |
| Appropriate place to sleep for infants | separate bed/crib | 24 | 4.2 |
| | The same bed with parents | 484 | 85.1 |
| | I don't know | 61 | 10.7 |
| Pillow under the mattress preferred for infants | Yes | 517 | 90.9 |
| | No | 52 | 9.1 |
| SIDS prevented by safe sleep position | Yes | 472 | 83 |
| | No | 97 | 17 |
| What do think exclusive breast feeding mean | feeding only breast milk for six months | 558 | 98.1 |
| | Feeding breast milk and complimentary food in combination | 11 | 1.9 |
| Maternal sleep position Knowledge | Good | 336 | 59.1 |
| | Poor | 233 | 40.9 |

**Table 5. Assessment of mother's infant sleep position practice (n = 569).**

| | | | |
|---|---|---|---|
| Infant sleeps in a crib/cott? | Yes | 24 | 4.2 |
| | No | 545 | 95.8 |
| Infant sleeps in a separate room from the caregiver? | Yes | 20 | 3.5 |
| | No | 549 | 96.5 |
| Is an infant Sharing the bed with the caregiver? | Yes | 53 | 93.7 |
| | No | 36 | 6.3 |
| Was the infant exposed to tobacco smoke during pregnancy or after birth? | Yes | 31 | 5.4 |
| | No | 538 | 94.6 |
| Does the infant sleep in a position other than supine (on the back)? | Yes | 561 | 98.6 |
| | No | 8 | 1.4 |
| How does the infant usually sleep? | Prone | 7 | 1.2 |
| | Back | 243 | 42.7 |
| | Side | 319 | 56.1 |

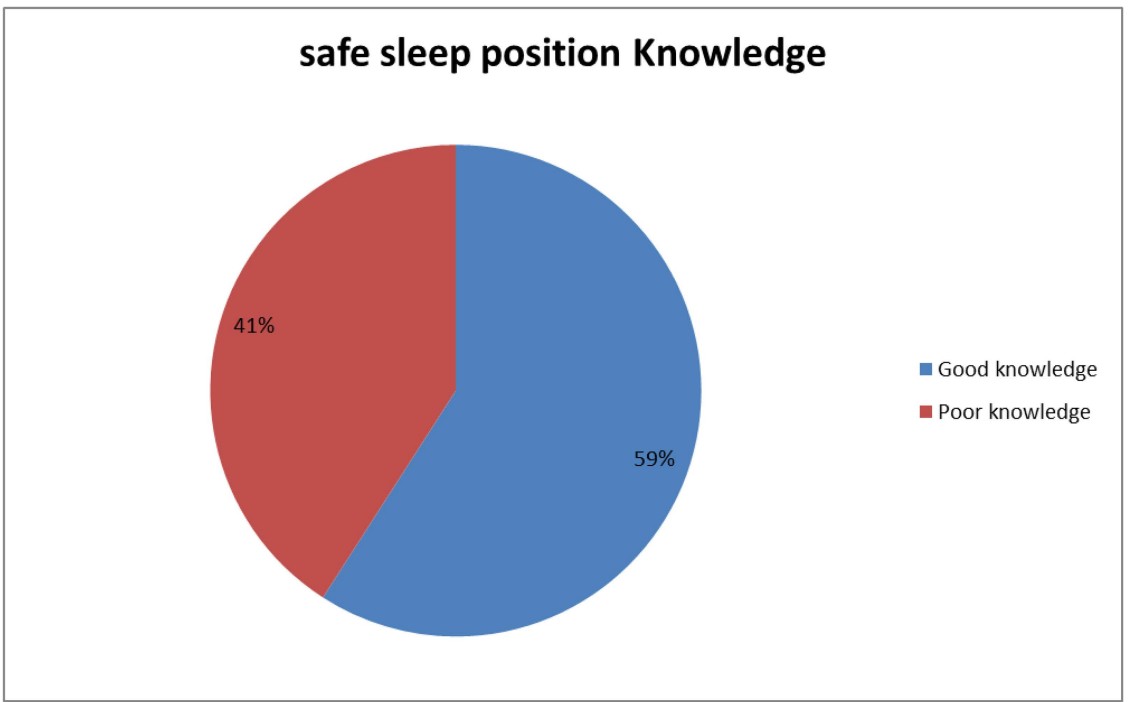

**Fig 2. Safe sleep position knowledge among mothers attending mother and child health service in Wolaita Zone public health institutions, Ethiopia, 2023.**

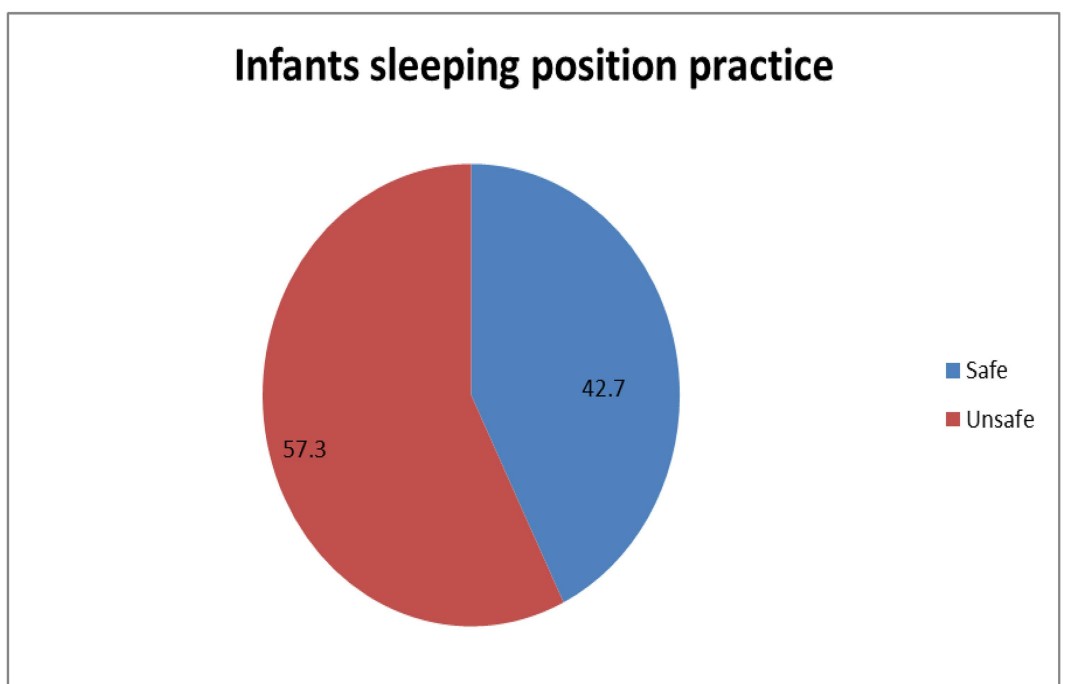

**Fig 3. Safe sleep position practice among mothers attending mother and child health service in Wolaita zone public health institutions, Ethiopia, 2023.**

**Table 6. Bivariate and multivariable analysis of factors associated with safe infant sleep position practice among mothers in public hospitals of Wolaita Zone, South Ethiopia, March – April 2023 (n = 569).**

| Variables | Category | Sleep position practice | | 95% CI | | p.value |
|---|---|---|---|---|---|---|
| | | Safe | Unsafe | COR | AOR | |
| Maternal age in years | ≤18 | 25 (49.0) | 26 (51.0) | 1.73(.86, 3.44)* | 2.27 (0.71, 7.28) | 0.167 |
| | 19-24 | 42 (30.0) | 98 (70.0) | 0.77 (0.44, 1.34) | 0.71 (0.27, 1.87) | 0.486 |
| | 25-30 | 69 (44.5) | 86 (55.5) | 1.44 (0.85, 2.44)* | 1.11 (0.48, 2.54) | 0.809 |
| | 31-36 | 73 (57.0) | 55 (43.0) | 2.38 (1.38, 4.11)* | 1.58 (0.76, 3.30) | 0.223 |
| | ≥37 | 34 (35.8) | 61 (64.2) | 1 | 1 | |
| Residence | Rural | 85(31.7) | 183 (68.3) | 1 | 1 | |
| | Urban | 158(52.5) | 143 (47.5) | 2.38 (1.689 3.351)* | 1.49 (0.93, 2.37 | 0.095 |
| Maternal Occupation | Employed | 91(75.2) | 30(24.8) | 8.90 (5.38, 14.73)* | **3.49 (1.81, 6.76)** | **<0.001** |
| | Unemployed | 1(11.1) | 8 (88.9) | 0.37 (0.045, 2.99) | 0.39 (0.05, 3.34) | 0.391 |
| | Self employed | 71(44.9) | 87 (53.1) | 2.39 (1.57, 3.67)* | **1.75 (1.05, 2.91)** | **0.031** |
| | Student | 18(50.0) | 18 (50.0) | 2.94 (1.44, 5.99)* | 1.81(0.79, 4.13) | 0.158 |
| | House wife | 62(25.4) | 182(74.6) | 1 | 1 | |
| Family size | 1 - 4 | 156(43.1) | 206(56.9) | 4.92 (2.03, 11.92)* | 2.16(0.64, 7.29) | 0.215 |
| | 5 - 8 | 81 (50.0) | 81(50.0) | 6.50 (2.61, 16.20)* | 2.41(0.79, 7.32) | 0.121 |
| | > 8 | 6 (13.3) | 39 (86.7) | 1 | 1 | |
| Parity | Primipara | 95(45.7) | 113(54.3) | 0.89 (0.60, 1.32) | **0.39 (0.21, 0.74)** | **0.004** |
| | Multipara | 56(32.6) | 116(67.4) | 0.51 (0.33, 0.78)* | **0.31 (0.13, 0.74)** | **0.008** |
| | Grand multipara | 92(48.7) | 97(51.3) | 1 | 1 | |
| ANC follow ups during last pregnancy | Yes | 233(44.4) | 292(55.6) | 0.37 (0.178, 0.762)* | 1.73 (0.69, 4.36) | 0.243 |
| | No | 11 (25.0) | 33(75.0) | 1 | 1 | |
| Number of ANC visit | No | 10 (22.7) | 34 (77.3) | 1 | 1` | |
| | 1 - 3 | 48(32.2) | 101 (67.8) | 0.30(0.15,0.63)* | 1.10 (0.66, 0.83) | 0.706 |
| | 4 | 185(49.2) | 191 (50.8) | 0.49,(0.33,0.73)* | | 0.706 |
| Source of information about SP | My mother | 13(48.1) | 14 (51.9) | 2.66 (1.20, 5.89)* | 0.78(0.22, 2.78) | 0.695 |
| | Mother in-law | 8(34.8) | 15 (65.2) | 1.53 (0.63, 3.74) | 0.92(.34, 2.52) | 0.875 |
| | Grand mothers | 24 (41.4) | 34 (58..6) | 2.03 (1.14, 3.61)* | 0.73(0.29, 1.82) | 0.505 |
| | Health professionals | 114 (83.8) | 22(16.2) | 14.87(8.84, 24.99)* | **4.64 (1.77,12.18)** | **<0.001** |
| | I didn't hear | 84(25.8) | 241(74.2) | 1 | 1 | |
| SIDS information | Yes | 176(45.8) | 208 (54.2) | 1.83 (1.27, 2.64)* | 0.64(0.37, 1.12) | 0.116 |
| | No | 178(46.4) | 206 (53.6) | 1 | 1 | |
| Sleep practice knowledge | Good knowledge | 190 (56.5) | 146 (44.5) | 4.42(3.04, 6.43)* | **2.04 (1.22, 3.40)** | **0.006** |
| | Poor knowledge | 53(22.7) | 180(77.3) | 1 | 1 | |

*Statistical significance in **COR**, **Statistical significance in **AOR**, **CI** = confidence interval, **COR** = crude odds ratio, **AOR** = adjusted odds ratio, **ANC** = Antenatal care, **SP** = Sleep position

Nigeria [23]. This relationship underscores the importance of maternal education and awareness in promoting infant safety. When mothers are informed about the risks and benefits of different sleep positions, they are more likely to make decisions that align with safe sleep guidelines [38].

### Strengths and limitations

This study has several notable strengths. First, the sample size was relatively large, and the number of outcome events (mothers practicing safe sleep) was sufficient to meet the minimum recommended events-per-variable (EPV) rule for

logistic regression analysis, which enhances the statistical power and validity of the multivariable model. Second, the study included participants from multiple public hospitals in different settings, which improves the representativeness of the findings within the region.

However, the study also has limitations. The cross-sectional design precludes the establishment of causal relationships between the independent variables and safe sleep practices. Data were collected through self-report, which may be subject to social desirability bias, potentially inflating the reported prevalence of recommended sleep practices. In addition, the exclusion of mothers who were healthcare workers and infants brought by non-caregivers (e.g., fathers or grandparents) may have introduced selection bias, as these groups may have different knowledge levels and behaviors. Finally, since the study was conducted in urban and semi-urban public hospitals, the findings may not be generalizable to rural populations or those accessing private or community-based health services.

## Conclusions

The safe infant sleeping position practice was relatively low compared to most previous studies and international recommendations. More than half of study participants were not practicing proper sleep position. Having governmental occupation, being self-employed, parity, source of information about sleep position, and having good knowledge about sleep position were significantly associated with safe sleep practice. By considering this, efforts should be made to ensure that accurate and evidence-based information regarding safe sleep practices is easily accessible to mothers through various channels, such as prenatal care visits and antenatal catchments. Furthermore, the ministry of health, hospitals, and health professionals should give attention to the application of safe sleep position practices that potentially reduce the risk of sleep-related infant deaths.

### Recommendations

#### For Researchers and Planners:

- Conduct longitudinal studies to assess the long-term effects of safe infant sleep practices on health outcomes, including the risk of sudden infant death syndrome (SIDS).

- Explore the effectiveness of interventions aimed at improving maternal knowledge and behavior regarding safe sleep.

- Advocate for the development and implementation of national policies that promote safe infant sleep, and collaborate with governmental and healthcare institutions to establish evidence-based guidelines.

#### For Hospitals:

- Provide regular training sessions for healthcare professionals—including doctors, nurses, and midwives—on current safe sleep recommendations.

- Develop and implement standardized institutional protocols outlining safe sleep practices and ensure consistent counseling of mothers during hospital visits.

- Display visual aids or posters in maternity and neonatal units to reinforce key messages about safe sleep positioning.

#### For Healthcare Providers:

- Use antenatal and postnatal care visits as opportunities to educate mothers about safe sleep positioning.

- Integrate counseling on infant sleep safety into routine maternal and child health services.

- Implement community-based health education programs to raise awareness among mothers and caregivers, especially in underserved areas.

## Supporting information

**S1 File. Questionnaire Amharic version.**
(DOCX)

**S2 File. Questionnaire English version.**
(DOCX)

**S3 File. Questionnaire wolaytic version.**
(DOCX)

## Author contributions

**Conceptualization:** Tamene Demissie Lachore, Tamirat Toma Bekele, Alemayehu Beharu Tekle.

**Data curation:** Tamene Demissie Lachore, Tamirat Toma Bekele, Nega Degefa, Genet Assefa.

**Formal analysis:** Tamene Demissie Lachore, Tamirat Toma Bekele.

**Investigation:** Tamene Demissie Lachore, Tamirat Toma Bekele, Alemayehu Beharu Tekle, Nega Degefa, Genet Assefa.

**Methodology:** Tamene Demissie Lachore, Tamirat Toma Bekele.

**Supervision:** Nega Degefa, Genet Assefa.

**Visualization:** Tamene Demissie Lachore, Tamirat Toma Bekele.

**Writing – original draft:** Tamene Demissie Lachore, Tamirat Toma Bekele.

**Writing – review & editing:** Tamene Demissie Lachore, Tamirat Toma Bekele, Alemayehu Beharu Tekle, Nega Degefa, Genet Assefa.

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
