## [Decision Letter · Decision Letter 0]

8 Apr 2025

Dear Dr. Tekle,

Thank you for submitting your manuscript to PLOS ONE. After careful consideration, we feel that it has merit but does not fully meet PLOS ONE’s publication criteria as it currently stands. Therefore, we invite you to submit a revised version of the manuscript that addresses the points raised during the review process.

We look forward to receiving your revised manuscript.

Kind regards,

Abayeneh Girma

Academic Editor

PLOS ONE

Journal Requirements:

3. Please remove all personal information, ensure that the data shared are in accordance with participant consent, and re-upload a fully anonymized data set.

Additional guidance on preparing raw data for publication can be found in our Data Policy (https://journals.plos.org/plosone/s/data-availability#loc-human-research-participant-data-and-other-sensitive-data) and in the following article:http://www.bmj.com/content/340/bmj.c181.long.

4. Please include captions for your Supporting Information files at the end of your manuscript, and update any in-text citations to match accordingly. Please see our Supporting Information guidelines for more information:http://journals.plos.org/plosone/s/supporting-information.

Additional Editor Comments :

Academic editor comments to the author

• Some of your discussions are ambitious; as such, they are not sufficiently supported by research or study across the globe. Again, the justification in the discussion (why does your study vary or be consistent with other similar studies?) is also not supported by the study.

• There are editorial problems throughout the manuscript.

• There are many grammatical errors throughout the manuscript. The manuscript needs extensive review by the author, plus it should also be checked by a professional speaker of the English language (the article is not presented in an intelligible fashion and is written in Standard English).

• After mentioning the full form for any phrase the first time, then you should use the abbreviated one in other parts of your manuscript.

• Finally, please check all references to ensure that none of the cited articles have been retracted. You can use the Retraction Watch database, available here (http://retractiondatabase.org/)

Reviewers' comments:

Reviewer's Responses to Questions

**Comments to the Author**

1. Is the manuscript technically sound, and do the data support the conclusions?

Reviewer #1: Yes

Reviewer #2: Yes

2. Has the statistical analysis been performed appropriately and rigorously?

Reviewer #1: Yes

Reviewer #2: Yes

3. Have the authors made all data underlying the findings in their manuscript fully available?

Reviewer #1: Yes

Reviewer #2: Yes

4. Is the manuscript presented in an intelligible fashion and written in standard English?

Reviewer #1: No

Reviewer #2: Yes

Reviewer #1: I appreciate the review invitation from the editor. I am also grateful to the authors for their efforts. I remarked on comments and suggestions that I thought could improve the integrity and readability of the author's work. Please find them below.

Title:

• Please, make the title more informative.

Introduction

• This section lack coherence and logical organization of sentences and paragraphs. I suggest the authors to rewrite the introduction.

• The authors did not cite source appropriately. They would cite sources immediately after each sentences rather than citing all sources together at the end paragraphs. In addition, the citation in a sentence “Placing infants on their backs to sleep has become the recommended sleep position by major health organizations, including the AAP, the Centers for Disease Control and Prevention (CDC), and the World Health Organization (WHO)[7, 8].” seems incomplete.

• Authors did not show the magnitude and severity of the problem. For instance, they simply report the occurrence high infant mortality. They would show how many of infant mortality are accounted to unsafe sleeping position, including SIDS.

• There are also nonspecific paragraphs.

Method

• Would not you think that the exclusion of “Infants presented to clinic by non-caregivers/not upbringing the infant. And Infants whose mothers are health care workers.” cause sampling bias?

• “Three hospitals were selected by random sampling method, and then proportional allocation was done for each hospital in terms of the previous year the same monthly report number of child care service to data collection period” Was it the previous year the same monthly report of the same age group service or general childcare?

• As you executed proportional allocation, you would put the Kth value used for each hospital.

• Is the tool self-developed or adapted from previous literature? Is it validated?

• At which hospital unit you contacted the mothers of infants? How did you managed the emotions of mothers of severely sick infants? How many minutes did the interview took?

• “Respondents who answer greater than the mean score of questions provide for them regarding knowledge will be considered as having good knowledge. While those answers below the mean score are those with poor knowledge.” This sentence is duplicated.

• How did you score Knowledge about SIDS?

• How did you manage the consent signing process for undedicated mothers?

Results

• What is your reference for covariates?

• Put the income category of the international currency ($) corresponding to that of the local during the study period too.

• I doubt the credibility of your data. Participants who attended higher education outweigh the other categories. This finding is contrary to national data (https://www.weforum.org/reports/ab6795a1-960c-42b2-b3d5-587eccda6023). Would you justify it, please?

• Please, convert interrogatives in the tables to variable form. Avoid the terms “you” and “your”.

• “While the other 326 (57.3%) of respondents reported that they had not practiced the safe infant sleep position.” This statement is not necessary.

• It’s better to minimize the decimals of AOR to 2 digits or follow the journal’s guideline.

• I suggest removing the p-value of the COR or using an asterisk instead. It congests the table.

Discussion:

• “The present study is distinguished by its notable strengths, primarily pertaining to the utilization of a large sample size and the establishment of diverse settings.” As you have used a logistic regression analysis to test multiple variables, I think this is not convincing elaboration. Read the rule of thumb for sample size in logistic regression” for your reference.

• Conclusion:

• The recommendation section is duplicated.

General suggestions:

Please, correct grammar, punctuation, and tense errors.

Reviewer #2: The study addresses an important public health issue—safe infant sleep practices—in a region where such data are scarce. The topic is highly relevant, given the global emphasis on reducing Sudden Unexpected Infant Death (SUID) and the lack of localized evidence in Ethiopia. The manuscript is well-structured, but some areas require clarification, refinement, or correction to enhance its quality and impact.

**Do you want your identity to be public for this peer review?** For information about this choice, including consent withdrawal, please see our Privacy Policy

Reviewer #1: No

Reviewer #2: **Yes:**  Indiris Abdu Yimam

---

## [Author Response · Author response to Decision Letter 1]

1 May 2025

We have tried to address to the comments raised by the reviewers and the editors on a separate document submitted as 'Response to Reviewers'.

---

## [Decision Letter · Decision Letter 1]

1 Aug 2025

Dear Dr. Tekle,

Thank you for submitting your manuscript to PLOS ONE. After careful consideration, we feel that it has merit but does not fully meet PLOS ONE’s publication criteria as it currently stands. Therefore, we invite you to submit a revised version of the manuscript that addresses the points raised during the review process.

We look forward to receiving your revised manuscript.

Kind regards,

Ilker Kacer, Assoc. Prof. M.D.

Academic Editor

PLOS ONE

Journal Requirements:

Reviewers' comments:

Reviewer's Responses to Questions

**Comments to the Author**

Reviewer #3: All comments have been addressed

Reviewer #4: (No Response)

Reviewer #5: (No Response)

2. Is the manuscript technically sound, and do the data support the conclusions?

Reviewer #3: Yes

Reviewer #4: Yes

Reviewer #5: Yes

3. Has the statistical analysis been performed appropriately and rigorously?

Reviewer #3: Yes

Reviewer #4: Yes

Reviewer #5: Yes

4. Have the authors made all data underlying the findings in their manuscript fully available?

Reviewer #3: Yes

Reviewer #4: Yes

Reviewer #5: Yes

5. Is the manuscript presented in an intelligible fashion and written in standard English?

Reviewer #3: No

Reviewer #4: Yes

Reviewer #5: Yes

Reviewer #3: the study address an important public health concern and meets many of criteria for publication, however there are minor few concerns which if could be addressed the it is a fit for publication. the following are the recommendations;

1. there are some Grammatical errors in the paper for example instead of writing 'mother in law" it was written "Mother in low". these sentences should be reviewed and corrected

2. some of the terms were not clearly defined and these included in the 10 knowledge assessment items, the actual list should be provided and clearly defined

3. ensure all figures e.g. Fig 2 and Fig 3 are embedded with clear titles and legends. currently, they are referenced but seem separate

Reviewer #4: The manuscript presents a valuable study on maternal knowledge and practices concerning infant sleep positions in South Ethiopia. While the research addresses an important public health issue, it would benefit from clearer justifications for certain methodological choices. Overall, the study contributes to the understanding of safe infant sleep practices and offers actionable recommendations for improving maternal education and infant safety.

The manuscript is well-written and well-structured. However, it may benefit from few improvements.

• A systematic sampling method was used, but exclusions (e.g., mothers who are healthcare workers) may introduce bias. The rationale for these exclusions should be better justified.

• The process, including informed consent, is well-described, although the handling of illiterate participants could be more detailed.

• The manuscript mentions checking multicollinearity using VIF, but a discussion of how it was addressed or managed would enhance transparency.

Reviewer #5: 1. Scientific Merit

This study addresses an important yet underexplored public health issue by investigating fecal-oral parasitic transmission in public toilets. The research question is clearly defined, and the results are consistent with the data presented. This contribution is relevant both locally and globally.

2. Methodology and Statistical Analysis

The sampling methods and laboratory procedures are sufficiently described. However, the statistical reporting could be improved. The specific tests used, software applied, and significance thresholds should be mentioned. Tables should include p-values and footnotes indicating statistical significance to enhance the transparency.

3. Language and Structure

The manuscript is generally well written, but several long and complex sentences—especially in the Introduction and Discussion—impair clarity. A more concise and direct writing style is recommended. The discussion would benefit from a clearer articulation of the study implications for public health policy.

4. Ethics and Transparency

Although the ethics committee is named, the approval number is missing. Please include the approval number or provide a justification if the study was exempt from the approval. Ethical transparency is essential for publications.

5. Keywords and Literature Currency

The current keywords are overly broad in scope. Additionally, the inclusion of 1–2 more recent references (2022–2024) would enhance the study relevance. Please consider revising them to include specific terms such as 'fecal-oral transmission', 'Giardia lamblia', and 'public health surveillance.’

**Do you want your identity to be public for this peer review?** For information about this choice, including consent withdrawal, please see our Privacy Policy

Reviewer #3: **Yes:**  Shabani Ally Massawe

Reviewer #4: No

Reviewer #5: **Yes:**  Emine ÖZDEMİR KAÇER

---

## [Author Response · Author response to Decision Letter 2]

30 Aug 2025

Point-by-Point Response to Reviewers and Academic Editor

Dr. Ilker Kacer, Assoc. Prof. M.D.

[em.pone.0.95206d.724294ad@editorialmanager.com]

RE: Revised Submission – Manuscript ID: PONE-D-25-10950R1

Title: Maternal Knowledge and Practice of Safe Infant Sleep Position in South Ethiopia: Implications for Preventing Sleep-Related Infant Deaths

Dear Dr. Kacer,

We sincerely thank you and the reviewers for your thoughtful and constructive comments on our manuscript. We have carefully revised the manuscript in accordance with the feedback provided and believe that the revised version is substantially improved.

We would also like to respectfully bring to your attention that all comments provided by Reviewer #5 appear to relate to a different manuscript, focusing on fecal-oral parasitic transmission in public toilets. As these comments are not applicable to our study on safe infant sleep practices, we have noted this in our response letter. However, if any part of Reviewer #5’s feedback was intended for our submission, we are happy to provide clarification or additional revisions as needed.

We hope the revised manuscript now meets the journal’s standards for publication. Thank you again for the opportunity to revise and resubmit our work to PLOS ONE. We greatly appreciate your time and consideration.

Sincerely,

Dr. Alemayehu B. Tekle, MD, ECCM

Corresponding Author

School of Medicine, Wolaita Sodo University, Ethiopia

Email: alemayehub.tekle@wsu.edu.et

Reviewer #3

Comment 1: There are some grammatical errors in the paper, e.g., “Mother in low” instead of “Mother-in-law.”

Response: Thank you for pointing this out. We have corrected the phrase “Mother in low” to “Mother-in-law” and thoroughly reviewed the manuscript to fix additional typographical and grammatical errors.

Comment 2: Some of the terms were not clearly defined, including the 10 knowledge assessment items. The actual list should be provided and clearly defined.

Response: We appreciate this suggestion. We have now included a description of the 10 knowledge assessment items in the Operational Definitions section. These items are derived from established literature and assess maternal understanding of sleep position safety. The questionnaire is also provided as Supporting Information.

Comment 3: Ensure all figures, e.g., Fig 2 and Fig 3, are embedded with clear titles and legends. Currently, they are referenced but seem separate.

Response: Thank you for this helpful comment. We have now ensured that all figures (including Figure 2 and Figure 3) are embedded in the manuscript with clear and descriptive titles and legends. While the figures will also be submitted separately in accordance with journal guidelines, we have revised the manuscript to improve clarity and ease of interpretation.

Reviewer #4

Comment 1: A systematic sampling method was used, but exclusions (e.g., healthcare worker mothers) may introduce bias. Please justify.

Response: Thank you for your observation. We have added justification for the exclusion of healthcare worker mothers. Mothers who are healthcare professionals were excluded to avoid potential bias due to their advanced medical knowledge, which may not reflect the general population's awareness and behavior.

Comment 2: The informed consent process for illiterate participants needs more detail.

Response: We agree. We have revised the Ethical Consideration section to describe how verbal consent and thumbprints were used for illiterate participants, ensuring ethical compliance and participant autonomy.

Comment 3: The manuscript mentions checking multicollinearity using VIF but doesn’t explain how it was addressed.

Response: Thank you. We now state that multicollinearity was assessed using VIF, and no significant multicollinearity was detected (maximum VIF = 2.9). This is now clarified in the Data Analysis section.

Reviewer #5

General Comments:

We sincerely appreciate Reviewer #5’s time and effort in providing feedback. However, upon careful review, we believe that all of the comments provided (e.g., references to fecal-oral parasitic transmission, public toilets, and related keywords such as Giardia lamblia) pertain to a different manuscript and are not relevant to our study, which focuses on maternal knowledge and practice of safe infant sleep position in South Ethiopia.

Should any of these comments have been intended for our manuscript, we would be grateful for clarification and are happy to respond accordingly.

---

## [Decision Letter · Decision Letter 2]

7 Oct 2025

Dear Dr. Tekle,

Thank you for submitting your manuscript to PLOS ONE. After careful consideration, we feel that it has merit but does not fully meet PLOS ONE’s publication criteria as it currently stands. Therefore, we invite you to submit a revised version of the manuscript that addresses the points raised during the review process.

Please submit your revised manuscript by Nov 21 2025 11:59PM. If you will need more time than this to complete your revisions, please reply to this message or contact the journal office at plosone@plos.org . A rebuttal letter that responds to each point raised by the academic editor and reviewer(s). You should upload this letter as a separate file labeled 'Response to Reviewers'.A marked-up copy of your manuscript that highlights changes made to the original version. You should upload this as a separate file labeled 'Revised Manuscript with Track Changes'.An unmarked version of your revised paper without tracked changes. You should upload this as a separate file labeled 'Manuscript'.

We look forward to receiving your revised manuscript.

Kind regards,

Ilker Kacer, Assoc. Prof. M.D.

Academic Editor

PLOS ONE

Journal Requirements:

Reviewers' comments:

Reviewer's Responses to Questions

**Comments to the Author**

Reviewer #6: All comments have been addressed

Reviewer #7: All comments have been addressed

Reviewer #8: (No Response)

2. Is the manuscript technically sound, and do the data support the conclusions?

Reviewer #6: Yes

Reviewer #7: Yes

Reviewer #8: Yes

3. Has the statistical analysis been performed appropriately and rigorously?

Reviewer #6: Yes

Reviewer #7: Yes

Reviewer #8: Yes

4. Have the authors made all data underlying the findings in their manuscript fully available?

Reviewer #6: Yes

Reviewer #7: Yes

Reviewer #8: Yes

5. Is the manuscript presented in an intelligible fashion and written in standard English?

Reviewer #6: Yes

Reviewer #7: Yes

Reviewer #8: Yes

Reviewer #6: Prevention and education of correct infant sleep position and the impact on the infant life is well addressed on the guidelines and studies. However, this study is consider important to assess the mother awareness programs and enhance prevention methods during antenatal and postnatal period to reduce the infant death.

Reviewer #7: The paper presents the results of simple questionairre-based study into the sleep position practices of a selection of South Ethiopian mothers attending a specific health service. A total of 569 study participants were included with a an unusually high response rate of 99.1%. The safe infant sleeping position practice (back to sleep) in this study

was 42.7 % [95% CI (38.6%, 46.8)]. The authors compared the results with several other countries.

Because breastfeeding has such a profound effect on SIDS rates (especially in combination with safe sleep position) it is a pity this was not also studied (see Fleming et al doi:10.1136/bmj.313.7051.191)

Reviewer #8: Background: Sudden infant death syndrome has been reduced globally over the past 30 years due to changes in recommendations, including supine sleep for infants. However, uptake of these recommendations has varied by country and little is known about the practice and knowledge in countries such as Ethiopia. These investigators conducted a cross-sectional study in the Wolaita Zone of Ethiopia to understand knowledge and practice of safe sleep recommendations for infants. A relatively large cohort was included with a very high response rate.

Comment

1. Abstract: not sure if there is a word limit, but could be shortened a bit if needed (especially Methods section). Background: should “sudden and unexpected death in infancy” be “sudden unexpected infant death” or “sudden infant death syndrome”? Would try to keep terms consistent for clarity if possible.

2. Introduction (page 2): first full paragraph (“Each year infants die…”) and second full paragraph (“An unsafe sleeping position…”) should probably be combined for clarity. for this audience, may want to distinguish the term SUDI from SIDS since they are being used together in this paragraph and may be confusing. “SIDS is nearly 2,500 in the United States”: is this deaths per year? Please clarify.

3. Introduction (page 3): Paragraph that starts with “Safe sleep position practice…” this paragraph seems out of place since the discussion has shifted from SIDS recommendations to the experience in Africa. Would suggest moving earlier in the Introduction.

4. Exclusion criteria (page 6): “Infants presented to…”: seems redundant since mother is needed for inclusion. “Congenital diseases”: maybe change to "conditions" and would clarify that these would potentially result in children being recommended to sleep in a different position.

5. Operational Definition and Measurement (page 11): “Respondents who scored the mean and above”: Would clarify if this was the mean from this cohort or a pre-set score (i.e. was the design so that exactly half would be scored as having "good knowledge?"

6. Sociodemographic characteristics of the respondents (page 14): is “housewives” the preferred term locally? Are they not currently working because they have an infant at home?

7. Family income (Table 2): is this per week/month?

8. Page 18 (paragraph at bottom of page): “Almost all infants 98.6% were sleep other than supine”: This sentence is confusing and is key to the manuscript. Would revise to clarify. Based on the table, seems like this is a combination of infants who primarily sleep on their back and maybe those who are able to roll onto their back from other positions?

9. Factors associatd with safe infant sleep position practice (page 21): “family income parity”: Does "parity" here refer to family income or the number of babies a mother has had? Please clarify.

10. Material occupation (Figure 6, page 22): would change p=0.000 to p<0.001 or similar.

11. Discussion (second paragraph, page 25): “The prevalence found in this study was lower…”: Was this asked in the same way ("how does the infant usually sleep?") in this cohort compared with those other studies? Older infants, even if initially placed in the supine position, may roll over, and not sure if this is part of the reason that number could be lower.

12. General: “data was” should be changed to “data were” through the manuscript.

**Do you want your identity to be public for this peer review?** For information about this choice, including consent withdrawal, please see our Privacy Policy

Reviewer #6: No

Reviewer #7: No

Reviewer #8: No

---

## [Author Response · Author response to Decision Letter 3]

15 Nov 2025

Dr. Ilker Kacer, Assoc. Prof. M.D.

[em.pone.0.969c88.cc76924f@editorialmanager.com]

RE: Revised Submission – Manuscript ID: PONE-D-25-10950R2

Title: Maternal Knowledge and Practice of Safe Infant Sleep Position in South Ethiopia: Implications for Preventing Sleep-Related Infant Deaths

Dear Dr. Kacer,

We are pleased to resubmit our revised manuscript entitled:

“Maternal Knowledge and Practice of Safe Infant Sleep Position in South Ethiopia: Implications for Preventing Sleep-Related Infant Deaths.”

We sincerely thank you and the reviewers for the constructive and insightful comments provided on our initial submission. We carefully considered all suggestions and have revised the manuscript accordingly to improve clarity, methodological transparency, and overall scientific quality.

We have included a detailed, point-by-point response document addressing every reviewer comment.

We believe that these revisions have substantially strengthened the manuscript, and we are hopeful that the revised manuscript will now be suitable for publication in PLOS ONE.

Sincerely,

Dr. Alemayehu B. Tekle, MD, ECCM

Corresponding Author

School of Medicine, Wolaita Sodo University, Ethiopia

Email: alemayehub.tekle@wsu.edu.et

---

## [Decision Letter · Decision Letter 3]

7 Dec 2025

Maternal Knowledge and Practice of Safe Infant Sleep Position in South Ethiopia: Implications for Preventing Sleep-Related Infant Deaths

PONE-D-25-10950R3

Dear Dr. Tekle,

We’re pleased to inform you that your manuscript has been judged scientifically suitable for publication and will be formally accepted for publication once it meets all outstanding technical requirements.

Kind regards,

Ilker Kacer, Assoc. Prof. M.D.

Academic Editor

PLOS One

Additional Editor Comments (optional):

Reviewers' comments:

Reviewer's Responses to Questions

**Comments to the Author**

Reviewer #6: All comments have been addressed

Reviewer #8: All comments have been addressed

2. Is the manuscript technically sound, and do the data support the conclusions?

Reviewer #6: Yes

Reviewer #8: Yes

3. Has the statistical analysis been performed appropriately and rigorously?

Reviewer #6: Yes

Reviewer #8: Yes

4. Have the authors made all data underlying the findings in their manuscript fully available?

Reviewer #6: Yes

Reviewer #8: Yes

5. Is the manuscript presented in an intelligible fashion and written in standard English?

Reviewer #6: Yes

Reviewer #8: Yes

Reviewer #6: Maternal Knowledge and Practice of Safe Infant Sleep Position in South Ethiopia:

Implications for Preventing Sleep-Related Infant Deaths. Its is important study for improving mother and community awareness and address infant care

Reviewer #8: Background: Sudden infant death syndrome has been reduced globally over the past 30 years due to changes in recommendations, including supine sleep for infants. However, uptake of these recommendations has varied by country and little is known about the practice and knowledge in countries such as Ethiopia. These investigators conducted a cross-sectional study in the Wolaita Zone of Ethiopia to understand knowledge and practice of safe sleep recommendations for infants. A relatively large cohort was included with a very high response rate.

All of my previous comments have been addressed with this revision.

The authors should be commended for their work on this project.

**Do you want your identity to be public for this peer review?** For information about this choice, including consent withdrawal, please see our Privacy Policy

Reviewer #6: No

Reviewer #8: No

---

## [Editor Report · Acceptance letter]

PONE-D-25-10950R3

PLOS One

Dear Dr. Tekle,

I'm pleased to inform you that your manuscript has been deemed suitable for publication in PLOS One. Congratulations! Your manuscript is now being handed over to our production team.

Kind regards,

on behalf of

Mr. Ilker Kacer

Academic Editor

PLOS One